



# Air-sea gas exchange measurements helped derive in-situ organic and inorganic carbon fixation in response to Ocean Alkalinity Enhancement in a temperate plankton community

Julieta Schneider[1], Ulf Riebesell[1], Charly A. Moras[2,3], Laura Marín-Samper[4], Leila R. Kittu[1], Joaquín Ortíz-Cortes[1], Kai G. Schulz[2]

[1] GEOMAR Helmholtz Centre for Ocean Research Kiel, Wischhofstrasse 1-3, 24148 Kiel, Germany
[2] Faculty of Science and Engineering, Southern Cross University, Lismore, NSW, Australia
[3] Institute for Geology, Universität Hamburg, 20146, Hamburg, Germany
[4] Instituto de Oceanografía y Cambio Global, Universidad de Las Palmas de Gran Canaria, 35017 Telde, Spain

Correspondence to: Julieta Schneider (jschneider@geomar.de) and Kai G. Schulz (kai.schulz@scu.edu.au)

**Abstract.** Ocean Alkalinity Enhancement (OAE) is a carbon dioxide removal strategy that aims to chemically sequester atmospheric $CO_2$ in the ocean while potentially alleviating localized effects of ocean acidification. Depending on the implementation approach, OAE can considerably alter seawater carbonate chemistry, resulting in reduced $CO_2$ partial pressures ($pCO_2$) and high pH. To investigate the effects of OAE on biogeochemical processes and organisms under close-to-natural conditions, a large-scale mesocosm experiment was conducted in the temperate fjord ecosystem near Bergen, Norway during late spring. A non-$CO_2$-equilibrated approach was chosen, simulating OAE with calcium- and silicon-based minerals. A gradient of five OAE levels was achieved by increasing total alkalinity (TA) by 0-600 μmol kg$^{-1}$. The added TA remained relatively stable over the 47-day experimental period and the measured $CO_2$ gas exchange rate was comparable to what would be expected for large oceanic regions. We estimated that full equilibration (95%) for a ΔTA of 600 μmol kg$^{-1}$ would take ~1050 days. Furthermore, there were a number of mineral-type and $pCO_2$/pH effects, with cumulative coccolithophore calcification showing an optimum curve response to decreasing $pCO_2$, consistent with findings from single-species laboratory cultures, while no mineral-type effect was observed. In contrast, *in-situ* net community production was higher in the silicate treatments but there was no $pCO_2$ effect. Zooplankton respiration, estimated from *in-situ* net community production and *in-vitro* net community production incubations, was lower for the silicate treatments and negatively correlated with $pCO_2$. These complex findings suggest both direct and indirect effects of mineral type and OAE level and provide a valuable foundation for designing future OAE field trials.



## 1 Introduction

The rapid increase in atmospheric carbon dioxide ($CO_2$) concentration over the last 250 years has been identified as a major cause for global warming, with current modelling projections for 2100 exceeding 2°C above pre-industrial levels (Peters, 2016; Rogelj et al., 2016). This is a critical threshold associated with the onset of extreme weather events (Schellnhuber et al., 2016). However, decreasing $CO_2$ emissions will not be sufficient to achieve a net-zero target towards the end of this century (Ho, 2023), which would be required to stay below the 2°C threshold (Rogelj et al., 2018). Therefore, it is paramount to explore the potential of active atmospheric $CO_2$ removal (CDR) strategies (IPCC, 2021; Van Vuuren et al., 2018). Ocean alkalinity enhancement (OAE) is a marine CDR approach with significant potential, that relies on speeding up the natural process of rock weathering via the addition of alkaline solutions/minerals. By adding alkaline feedstocks to seawater (Kheshgi, 1995), the subsequent increase in alkalinity and pH lowers surface water $CO_2$ concentrations, hence creating a $CO_2$ sink or reducing a $CO_2$ source if the water is naturally oversaturated (Caserini et al., 2021; Denman, 2008; Feng et al., 2016; Hartmann et al., 2013; Köhler et al., 2010; National Academies of Sciences, 2022; Sabine and Tanhua, 2010). The localized increase in total alkalinity (TA) and pH simultaneously mitigates ocean acidification (OA), even after eventual $CO_2$ equilibration with the atmosphere (Moras et al., 2022). Modelling studies suggest that the $CO_2$ uptake potential of OAE ranges between 14 and 41 gigatons per year (Oschlies et al., 2023). This variability underscores a significant knowledge gap between modelled predictions and real-world conditions, which needs to be acknowledged and addressed (Henderson et al., 2008).

There are two very distinct approaches to performing OAE additions: $CO_2$-equilibrated, which involves equilibrating the high TA seawater with atmospheric $CO_2$ prior to release, and non-$CO_2$-equilibrated, which relies on natural air-sea gas exchange to achieve equilibrium over time. For the latter process to happen, it is crucial that the alkalized water remains in the surface ocean in contact with the atmosphere. Since gas exchange could take months to years (He and Tyka, 2023) and depend heavily on the degree of dilution of the high-TA water with the surrounding seawater, marine organisms might be exposed to $CO_2$ depleted conditions and relatively high pH levels, which could be detrimental to planktonic communities (Doney et al., 2020; Kroeker et al., 2010). However, experimental data on potential biological OAE effects are scarce (National Academies of Sciences, 2022), and thresholds of applicability not yet fully understood. For instance, recent studies have identified the potential risks of runaway calcium carbonate ($CaCO_3$) precipitation beyond certain pH thresholds which should be avoided since it reduces the CDR potential of OAE (Fuhr et al., 2022; Hartmann et al., 2023; Moras et al., 2022; Paul et al.,2024; Suitner et al., 2024).

A number of potentially suitable OAE feedstocks have been previously suggested (Hartmann et al., 2013; Renforth and Henderson, 2017), including quick or hydrated lime (calcium-based), brucite (magnesium-based), and olivine (silicate-based), all of which release soluble products of interest for marine organisms (Montserrat et al., 2017; Moras et al., 2024). For these materials, Bach et al., (2019) hypothesized that the increase in calcium ions ($Ca^{+2}$) together with DIC upon equilibration, would benefit the calcification process of key calcifiers such as coccolithophores, which are highly impacted by ocean acidification; while the release of $Si(OH)_4$ could benefit diatoms, another group of primary producers that rely on silicate to synthesize their exoskeletons. This hypothesis, termed "*white vs green ocean*" stresses the importance to assess the biological response of natural ecosystems to OAE with respect to the type and concentration of the alkaline material used.

This research presents the first data on the temporal dynamics of the carbonate system within large-scale mesocosms following deployment of silicon- and calcium-based OAE treatments ranging from 0 to 600 µmol kg⁻¹ of added alkalinity. We conducted a 53-day *in situ* experiment using 10 pelagic mesocosms (Riebesell et al., 2013) in Bergen, Norway. OAE release was in a non-pre-equilibrated way, i.e., the ingassing of $CO_2$ was left to occur naturally via air-sea gas exchange, which is believed to represent the most feasible deployment scenario from both a technological and an economical



perspective (Schulz et al., 2023). However, this approach can also lead to strong perturbations of seawater carbonate chemistry, namely high pH, low partial pressure of seawater $CO_2$ ($pCO_2$), and high seawater aragonite saturation states ($\Omega_A$). These perturbations would be highest close to an OAE point source prior to dilution and atmospheric $CO_2$ uptake.

Here, we focused on assessing OAE induced carbonate chemistry changes, the stability of alkalinity over time (including biogenic calcification), as well as deriving air-sea $CO_2$ exchange rates from measurements, allowing to also calculate net ecosystem primary productivity. Processes that need to be better understood before large field deployments of OAE and for later monitoring and verification.

## 2 Materials and Methods

### 2.1 Mesocosm deployments and experimental setup

Ten Kiel Off-Shore Mesocosms for Future Ocean Simulations (KOSMOS) were deployed by the research vessel ALKOR and moored in the Raunefjord, Norway on the 7th of May 2022 (60.25N 5.2E). The technical design of these seagoing mesocosms and procedures for water column manipulation are described in detail by Riebesell et al (2013). Briefly, the 20 m long mesocosm bags were suspended in 8m tall floating frames and both ends were covered with a 3 mm mesh size

net to exclude larger organisms during filling. The tops of the bags were then submerged 1 meter below the sea surface and fully unfolded to enclose a waterbody containing the natural planktonic community. The bags were left submerged for 5 days to allow for sufficient seawater exchange to ensure similar starting conditions in all mesocosms. Next, the waterbody within the mesocosm bags was isolated from the surroundings by attaching a 2 m-long, funnel-shaped sediment trap on the lower end of the bags and lifting the top end 1m above the surface. The attachment of the sediment trap on the

13[th] of May marked the beginning (day 0) of the 53-day experiment. Right after closure, a 1mm net was pulled from bottom to top to remove any heterogeneously distributed nekton. From day 1 to 3, daily samplings were conducted to monitor the initial conditions of the enclosed waters before OAE manipulation on day 6. Additionally, a volume determination based on the salinity change after addition of a brine solution was carried out following Czerny et al. (2013), yielding an estimated average volume of ~$61.6 \pm 1.9$ m[3] in the mesocosms.

All water column manipulations in the mesocosms were achieved with the use of a pumped injection device equipped with polycarbonate piping of various lengths (Riebesell et al., 2013), that was lowered and raised several times in each mesocosm during manipulations to ensure homogeneous distribution throughout the entire water column.

### 2.2 OAE manipulation

The alkalinity manipulation was performed on day 6 by addition of a sodium hydroxide (NaOH, Merck) solution. To

100 simulate the use of the two alkaline feedstocks: hydrated lime ($Ca(OH)_2$) and olivine (as forsterite, $Mg_2SiO_4$), NaOH additions were followed by the addition of respective $Ca^{2+}$ and $Mg^{2+}$ rich solutions. These two solutions were prepared using reagent grade calcium chloride ($CaCl_2$) and magnesium chloride ($MgCl_2$). Furthermore, to simulate the release of $SiO_3^{2-}$, under the olivine scenario, a Si-rich solution was prepared using $Na_2SiO_3$ (Roth) and added in equal concentrations to all Si-treatments (75 µmol/L), regardless of the targeted alkalinity increase (more details can be found in (Goldenberg

et al., 2024). The reasoning behind this decision was to avoid colloid formation that occurs at high Si concentrations (up to 150 µmol/L would have to be added to the highest TA treatment to match the TA to silicate ratio of 4:1 in olivine), and to allow separating between silicate and TA effects. All solutions were prepared by dissolving the corresponding salts in 20L deionized water (Milli-Q®, 18.2 ΩM). Each salt was weighed and dissolved separately. In summary, stock solutions of NaOH and $CaCl_2$ were added to the Ca-based treatments, and $MgCl_2$, NaOH and $Na_2SiO_3$ to the Si-based ones (the

unavoidable increase of TA by adding silicate in the control and ΔTA 150 µmol kg[-1] was compensated for by adding HCl and not adding NaOH respectively).



### 2.3 N₂O spiking and nutrient addition

For air-sea gas exchange determination, nitrous oxide ($N_2O$) additions were performed on day 14 following the procedure described in Czerny et al. (2013). One liter of a saturated stock solution was prepared by bubbling 0.2 μm filtered seawater for two days with $N_2O$ (Nippon Gases). The amounts of stock solution to be added in each mesocosm were calculated using solubility constants by Weiss & Price (1980) and taking into consideration in situ salinities, temperatures and individual mesocosm volumes. The stock solution was then diluted with filtered seawater into 25 L carboys and each mesocosm was spiked with one carboy.

Given the duration of the experiment and the low levels of inorganic nutrients in comparison to the surrounding coastal water, a nutrient addition was performed on day 26. Nitrate concentrations ($NO_3^-$) were targeted at 4 μmol $L^{-1}$ and phosphate concentrations ($PO_4^{3-}$) were enhanced following the N:P Redfield ratio of 16:1 (Redfield, 1934).

Ca treatments also received a minor silicate ($Si(OH)_4$) addition in a N:Si ratio of 4:1 to better mimic the natural conditions, without creating a Si-enrichment scenario. Nutrient concentrations were measured the day before nutrient addition and ~2 hours after for quantification and to ensure targeted stoichiometry. It was then noted that the stoichiometry was not even across mesocosms due to underestimated nitrate additions. A successful nitrate amendment was performed on day 28.

### 2.4 Sampling procedures and CTD operations

Following the alkalinity manipulation, randomized sampling was carried out every second day in the morning hours (08:00 – 12:00). Depth-integrated (0-20m) water samples were taken from the mesocosms and the surrounding coastal water (later referred to as "Fjord") using a 5 L integrating water sampler (IWS, HYDRO-BIOS, Kiel).

Samples were collected in decreasing order of sensitivity to gas exchange, i.e., $N_2O$, followed by carbonate chemistry and inorganic nutrients. $N_2O$ samples were drawn with a Tygon tube from the IWS directly into 20mL caramel vials in triplicates. After making sure vials were bubble-free, they were crimp sealed immediately and kept at room temperature after fixation with 10 μL of a saturated mercury chloride solution ($HgCl_2$). To minimize gas exchange during air transport and storage, two paraffin wax coats were applied to the crimp seals (Glatzel and Well, 2008; Kock et al., 2016). For carbonate chemistry parameters, such as pH and TA, 0.5 L of seawater were taken into air-tight glass flasks. Clean bottles were pre-rinsed with sample water immediately prior to filling (using pre-rinsed Tygon tubing). Finally, to minimize air–water gas exchange, filling was done gently from bottom to top with an overflow of ~1.5 times the sampling volume (Dickson et al., 2007).

Nutrient subsamples were collected next into 250 mL acid cleaned polypropylene bottles. All samples were kept in cool conditions and protected from direct sunlight until further analysis. Subsamples for $NO_3^-$, $NO_2^-$, $PO_4^{3-}$, and $Si(OH)_4$ were filtered using a PES syringe filter (0.45 μm Sterivex, Merck) and analyzed spectrophotometrically following Hansen and Koroleff (1999).

CTD casts were performed with a multiparameter logging probe (CTD60M, Sea & Sun Technology) directly after the main sampling (14:00 - 16:00), yielding depth profiles of salinity, temperature and pH.

### 2.5 Sample Analysis

### 2.5.1 Carbonate Chemistry

All carbonate chemistry analyses were performed in the same way, starting by sterile-filtering the seawater samples through 0.2 μm syringe filters (Sartorius) using Tygon tubing connected to a peristaltic pump. The filtration process resulted in the removal of biomass and potential alkaline particles that may cause changes to seawater carbonate chemistry during analysis (Bockmon and Dickson, 2014). Water for gas-sensitive parameters was subsampled first with gentle flow



and providing an overflow ~1.5 times the final sampling volume. Samples were then kept at room temperature and measured within 12h.

TA was measured by open-cell potentiometric titration as described in Dickson et al. (2007). A 0.05M HCl solution with an ionic strength of 0.72 mol kg$^{-1}$ (corresponding to a salinity of 35 and adjusted by NaCl addition), was used as the titrant. Titrations were performed using a Metrohm Aquatrode Plus (Pt1000) connected to a 907 Titrando, with samples loaded onto an 862 Compact Titrosampler. The temperature was recorded during titration and varied between 20 and 25 °C, i.e., laboratory ambient temperature. 50 g of sample water were weighed into the titration beakers with a precision of 0.1 mg. For every run, results were corrected against certified reference material (CRM, batch 193, Dickson, 2010). Finally, TA was calculated using the titration curves and the "Calkulate" script within PyCO2SYS by (Humphreys et al., 2022). Each sample was measured in duplicate, and the corresponding average and standard deviations are reported. The TA measurement precision was calculated by error propagation of the samples and CRM standard deviations and averaged ±1.9 µmol kg$^{-1}$.

Seawater pH on the total scale (pH$_T$) was determined spectrophotometrically with a VARIAN Cary 100 in a 10 cm thermostated cuvette at 25°C using cresol purple as described in Dickson et al. (2007). Before measurement, samples were acclimated to 25°C in a thermostated water bath. To minimize potential CO$_2$ air–water gas exchange, a syringe pump (Tecan Cavro XLP) was used for sample and dye mixing and for cuvette injection (see Schulz et al. (2017)). A more detailed description of pH$_T$ corrections is found in Section 2.6.3. The average pH$_T$ precision was estimated at ±0.003 units.

DIC samples were taken only on day 9 of the experiment to cross-check the estimated DIC derived from TA and pH$_T$ measurements. Said samples were fixed with HgCl$_2$ for later analysis, conducted on an Automated Infra-Red Inorganic Carbon Analyzer (AIRICA, Marianda), connected to a LI-COR LI-7000 (Gafar and Schulz, 2018). The samples were analyzed in triplicates, and the instrument uncertainty was estimated at ± 1.5 µmol.kg$^{-1}$.

### 2.5.2 N$_2$O

Aquatic N$_2$O concentrations were measured via gas chromatography (GC) with electron capture detection (ECD) (Hewlett Packard 5890 II), using a headspace static equilibration procedure (precision of ±1.8 %). The GC was equipped with a 6'/1/8'' stainless steel column packed with a 5Å molecular sieve (W. R. Grace & CO) and operated at a constant oven temperature of 190 °C using a 95/5 argon-methane mixture (5.0, AirLiquide) as carrier gas. A 10 mL headspace was manually created in each sample vial with helium (5.0, AirLiquide), and the overflowing water was collected with a second syringe. Next, the vials were shaken vigorously for 20s and left to settle for 2 hours at room temperature. Subsamples of the equilibrated headspace were then injected into the sample loop of the GC. Certified gas mixtures of N$_2$O in artificial air (Deuste Steininger GmbH) with mixing ratios of 330 ± 0.2 and 994 ± 0.2 ppb as well as 6:3 and 4:5 dilutions with helium were used to construct daily calibration curves with a minimum of three data points within the sample concentration range.

N$_2$O concentrations were calculated according to Walter et al. (2006) using the solubility function of (Weiss and Price, (1980). The average precision, calculated as mean standard deviation from triplicate measurements, was 0.7 nM.

### 2.6 Data Analysis

The 53-day experiment was divided into three distinct phases. The pre-treatment phase (day 1 to 6), from here on referred to as phase 0 for simplicity, denotes the baseline state of the system before any alkalinity manipulation. This preliminary phase was an important step to confirm that the starting conditions in all mesocosms were similar. The reaction phase after OAE manipulation (days 7 to 28), phase I, corresponds to the period in-between alkalinity manipulation and the addition of dissolved inorganic nutrients. This post-treatment and pre-fertilization phase mainly consisted of the



immediate response of the system to an increase in alkalinity levels in a post-bloom scenario. The last phase after nutrient
addition is referred to as phase II. This post-fertilization period (days 29 to 53) is thought to capture the mid-term indirect

responses to alkalinity addition of a biologically active ecosystem state.

Throughout the experiment, two carbonate chemistry parameters (TA and $pH_T$) were measured and coupled to the corresponding salinity and temperature measurements (daily averages from CTD casts), and nutrient concentrations so the remaining variables of the carbonate system could be calculated (such as DIC, $\Omega_{Ar}$ and $pCO_2$). To do so, the software PyCO2SYS version 1.8.2 (Humphreys et al., 2024) was used with chosen constants for calculations as follows: K1 and

K2 for carbonic acid from (Sulpis et al., 2020), $KHSO_4$ from (Dickson, 1990), KHF from (Dickson and Riley, 1979), $[B]_T$
from Uppström, (1974)and the universal gas constant R.

### 2.6.1 CO₂ fluxes and net community production estimates

To calculate the daily air-sea $CO_2$ fluxes, $N_2O$ was used as a tracer following the approach described by Czerny et al. (2013) and using measured $N_2O$ to derive transfer velocities (Fig. S3). Fluxes across the water surface ($F_{N_2O}$) were then

calculated as follows:

$$F_{N_2O} = \frac{I_{w1} - I_{w2}}{A * \Delta t} , (1)$$

where $I_{w1}$ and $I_{w2}$ are the fitted bulk water $N_2O$ inventories at time $t_1$ and $t_2$, respectively; A is the surface area of the

mesocosms; and $\Delta t$ is the time difference between $t_1$ and $t_2$. From here, first $N_2O$, and then $CO_2$ transfer velocities were calculated (see Czerny et al. (2013) for details, and note the typo in there, where for the calculation of chemical enhancement the boundary thickness layer z should read 0.02 cm), as shown in Eq. (2), which then allowed to estimate daily $CO_2$ fluxes ($F_{CO2}$), according to Eq. (3).

$$k_{CO_2} = \frac{k_{N_2O}}{\sqrt{\frac{Sc_{CO_2}}{Sc_{N_2O}}}} , (2)$$

$$F_{CO_2} = k_{CO_2} * \left( C_{CO_2w} - C_{CO_2weq} \right) , (3)$$

where $k_{CO_2}$ is the transfer velocity of $CO_2$, $Sc_x$ are the corresponding Schmidt numbers, and $C_{CO2w}$ and $C_{CO2weq}$ are the

bulk-water $CO_2$ concentration and the calculated equilibrium concentration with the atmosphere, respectively. Atmospheric $pCO_2$ was estimated at 417 µatm (referenced to late spring 2022; Lan et al., 2025; NOAA/GML). Fluxes in this paper are shown as daily changes (mmol C m$^{-2}$ d$^{-1}$).

Previous mesocosm experiments (Czerny et al., 2013; Spilling et al., 2016) have already shown how $CO_2$ fluxes can be greatly affected by chemical enhancement due to hydration reactions of $CO_2$ and conversion to $HCO_3^-$ and $CO_3^{2-}$ in the

boundary layer, particularly under low turbulence conditions and high pH (Wanninkhof and Knox, 1996). Given the high concentration of $OH^-$ during this experiment, we applied the correction for chemical enhancement by Hoover and Berkshire (1969) with refitted hydration and hydroxylation rate constants by Schulz et al. (2006). The enhancement factor (α) was calculated following Eq. (4):

$$\alpha = \frac{\tau}{\left[ (\tau-1) + \frac{\tanh(Q*z)}{(Q*z)} \right]} , (4)$$



Where τ represents the chemical enhanced flux and Q (calculated as shown in Czerny et al (2023)) represents the enhanced flux, the hydration of $CO_2$ and the diffusion coefficient. The average boundary layer thickness z (cm ± std) was calculated to be 0.017 ± 0.003 cm and the overall enhancement ranged from 6% to 20% during this experiment.

### 2.6.2 Calcification and net community production estimates

Cumulative calcification rates (CALC) were estimated following Eq. (5). For this, salinity normalized TA changes (salinity 33 normalized) were calculated, and the uptake of nitrate, i.e., $NO_3^-$, and phosphate, i.e., $PO_4^{3-}$, – each of which increases TA by 1 mol per mol of nutrient uptake (Wolf-Gladrow et al., 2007) – was factored in. The cumulative sum of salinity-normalized and nutrient-uptake-corrected TA changes (divided by 2 due to the double contribution of $CO_3^{2-}$ to TA) was then used to gauge overall $CaCO_3$ production in each treatment. Note that the decrease in TA by the uptake of
growth-requiring conservative cations such as $Mg^{2+}$, $K^+$ and $Ca^{2+}$ (other than that used for calcification) was ignored as being much smaller than the effect of nitrate uptake (Wolf-Gladrow and Klaas, 2024).

$$CALC = -(\Delta TA_{33} + \Delta[NO_3^-]_{33} + \Delta[PO_4^{3-}]_{33}) * 0.5 , (5)$$

The $CaCO_3$ production potential (CCPP), i.e., the amount of $CaCO_3$ produced during coccolithophorid bloom (the main
calcifier in our experiments), was estimated following Gafar et al. (2018). *In situ* temperature, carbonate chemistry speciation and light conditions at each day for each treatment were considered. Then, normalized CCPP was compared to normalized cumulative $CaCO_3$ production (CALC).

It was also possible to estimate the biologically mediated change of net community production derived from the inorganic carbon fraction ($NCP_{DIC}$) by accounting for the impact of cumulative $CO_2$ air-sea exchange ($cCO_2$) on the cumulative
DIC consumption (salinity 33 normalized) and factoring in the formation of $CaCO_3$, as expressed in Eq. (6):

$$NCP_{DIC} = -(\Delta(DIC + cCO_2)_{33} - CALC) , (6)$$

Note that $cCO_2$ is added to DIC in the equation because it has a negative sign by definition. The following change in sign
is meant for comparison purposes, since production is often referred to as positive.

### 2.6.3 pH corrections

Calculations of $pH_T$ included corrections for changes due to dye addition. For that purpose, a batch of sterile filtered seawater (natural seawater filtered through a 0.2 um filter) was prepared and used to achieve 5 different levels of pH by additions of a 1M NaOH solution that would simulate the experimental increments in TA (steps of 0, 150, 300, 450 and
600 μmol kg⁻¹). For each level, $pH_T$ was measured for 5 additions of increasing dye addition, and the change in $pH_T$ per addition of dye was calculated. This allowed for the establishment of a calibration curve with a linear correlation between $pH_T$ level and $pH_T$ change due to dye addition (for details see Dickson et al. (2007).

The working range of m-cresol dye has been suggested to be one pH unit below and above the indicator's $pK_2$ (Hudson-Heck et al., 2021), i.e., about 7-9 at 25˚C, which should have covered our experimental range. However, dye impurities
can reduce the working range significantly (Schulz et al., 2023). And indeed, we found that above $pK_2$, there were increasing deviations of measured versus calculated $pH_T$ (from measured DIC and TA on day 9). Hence, a linear correction was applied for those measurements (Fig. S2).



## 3 Results

### 3.1 Alkalinity enhancement and carbonate system variability

During phase 0, conditions in all mesocosms were relatively similar, averaging $2213 \pm 3$ µmol $kg_{sw}^{-1}$ for measured TA, $8.109 \pm 0.004$ for $pH_T$, $2030 \pm 4$ µmol $kg_{sw}^{-1}$ for DIC, $339 \pm 3$ µatm for $pCO_2$, and $2.00 \pm 0.02$ for $\Omega_A$ (Fig. 1). After the manipulation on day 7, $pH_T$ increased to a maximum of 8.767, and $\Omega_A$ up to 7.66 in the two highest treatments, while $pCO_2$ decreased to 61. In contrast, DIC was hardly affected (Fig. 1 b-e).

Overall, TA remained relatively stable throughout the experiment, regardless of the phase and the treatment (Ca- or Si-
based), with a maximum variability of 19 µmol $kg^{-1}$. $\Omega_A$ remained stable throughout Phase I, but showed an increase ranging from ~0.20 to ~0.45 units in each treatment from day 35 onwards (Fig. 1 a, c).

In contrast, changes in $pH_T$ and DIC were observed throughout the experiment. After an initial drop in DIC in all meso-cosms, DIC changes were less consistent over time, with increments of up to 8-28 µmol $kg^{-1}$ in the treatments before the nutrient addition. About a week after the nutrient addition, DIC decreased in all mesocosms again, by about 23-51 µmol
$kg^{-1}$, coinciding with average Chl $a$ changes from $0.53 \pm 0.1$ µ $L^{-1}$ up to $3.4 \pm 0.9$ µ $L^{-1}$ during phase II (Fig. S1).

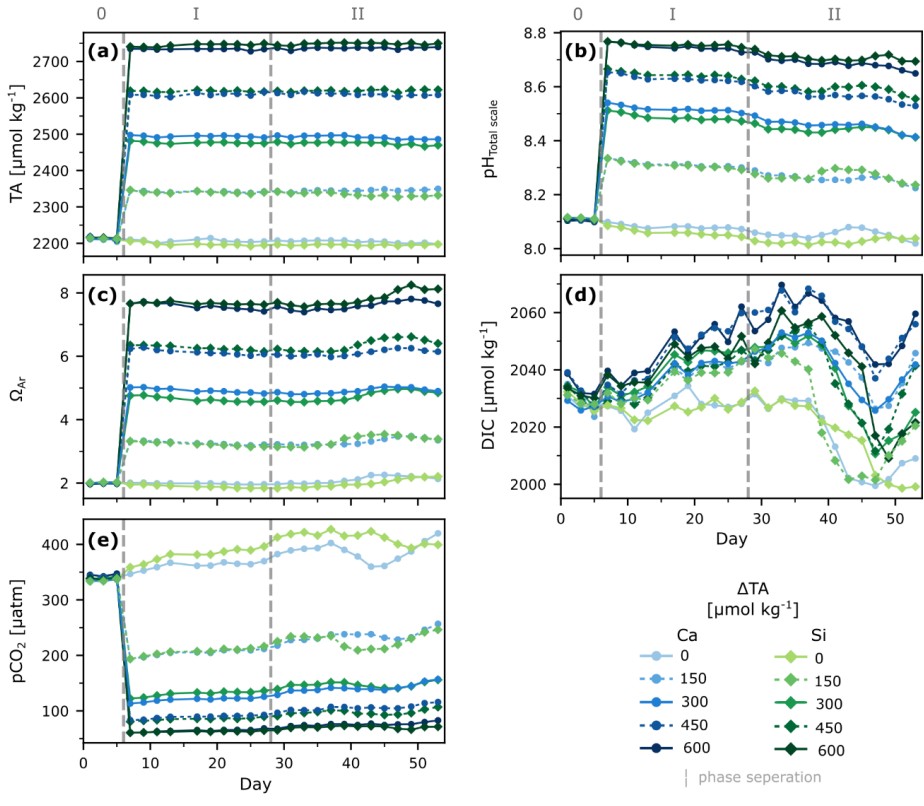

**Figure 1: Temporal development of carbonate chemistry parameters under different levels of non-CO₂-equilibrated OAE.**
Depth-integrated measured TA **(a)**, $pH_{Total\ scale}$ at in-situ temperatures **(b)**, and calculated aragonite saturation state **(c)**, dissolved inor-
ganic carbon **(d)**, and partial pressure of carbon dioxide **(e)**. Dashed lines and roman numbers denote the pre-treatment (0) phase and phases before (I) and after (II) nutrient addition.



### 3.2 Air-sea CO₂ gas exchange

The experiment began with initial $pCO_2$ below atmospheric levels, averaging $339 \pm 3$ µatm across treatments (Fig. 1). All treatments showed an immediate uptake of $CO_2$, leading to a daily increase in DIC of ~0.1 µmol kg$^{-1}$. While $F_{CO_2}$ didn't
change significantly for the control treatments right after the TA addition, it increased with increasing $\Delta$TA and corresponding lower seawater $pCO_2$ (Fig. 2). In the highest treatment, $F_{CO_2}$ was five times higher than in the controls. Interestingly, this increasing rate of $CO_2$ ingassing persisted in all treatments except for the controls, resulting in 50% increase towards the end of the experiment. The total cumulative net uptake of $CO_2$ ranged from ~16 µmol C kg$^{-1}$ to ~30 µmol C kg$^{-1}$ (see Fig. S4) across treatments. Notably, no differences in $F_{CO_2}$ were observed between the different minerals.

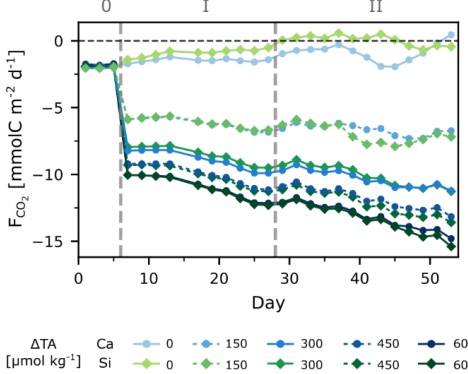


**Figure 2: Daily CO₂ fluxes over time**, with negative values indicating net influx (in-gassing), and positive values net outflux (outgassing) to the atmosphere. See section 2.6.1 for details. Dashed lines and roman numbers denote the different phases.

### 3.3 Estimated calcification rates and net community production

Except for the Ca-treatment $\Delta$TA 150, the response of normalized cumulative calcification to $pCO_2$ was that of an opti-
mum curve (fitting Eq. 9 from Gafar et al. (2018)), reaching a plateau at about 250 µatm (Fig. 3a). The highest calculated $CaCO_3$ production potential (CCPP) was observed at a $pCO_2$ of about 200 µatm (Fig. S5). With up to 12 µmol kg$^{-1}$ of $CaCO_3$ produced, cumulative calcification was highest in low-to-intermediate TA treatments and close to zero in the two highest ones (Fig. S4).

$NCP_{DIC}$ showed no significant differences between Si and Ca at the beginning (Fig. 4a). The community reacted to the
nutrient addition reaching a peak around 10 to 15 days after it. $NCP_{DIC}$ in the silicate treatments reached about 52 µmolC kg$^{-1}$, whereas in the calcium treatments the $NCP_{DIC}$ peaks were roughly half in magnitude. To better see potential differences during the bloom phase, we calculated the changes in daily NCP relative to the mean NCP values after nutrient addition and plotted the cumulative changes (Fig. 4b). This gives a clearer perspective of how differently the treatments reacted and the delays in time among blooms, if any, showing as a general trend that silicon-based treatments reached
higher bloom peaks than calcium-based ones, and higher alkalinity treatments took longer to start blooming. Though such a statement assumes that the sampling resolution was high enough to capture all real bloom peaks.

To better assess whether a mineral or treatment effect influenced the NCP responses, we proceeded to take maximum production value during the bloom for each treatment and plotted it against the mean $pCO_2$ value during the bloom period (Fig. 4c). An ANCOVA analysis showed no alkalinity effect, but a clear mineral effect on production.






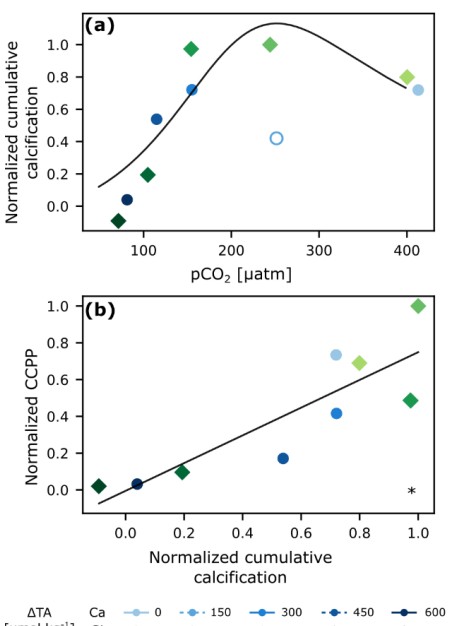

**Figure 3: Cumulative calcification** (average of last 2 days) derived from carbonate chemistry parameters vs $pCO_2$ with optimum curve fitting (Eq. 9 in Gafar et al. (2018)) (**a**), and normalized cumulative $CaCO_3$ production potential (CCPP) vs normalized cumulative calcification (**b**). The hollow circle was excluded from analysis. * $p<0.05$

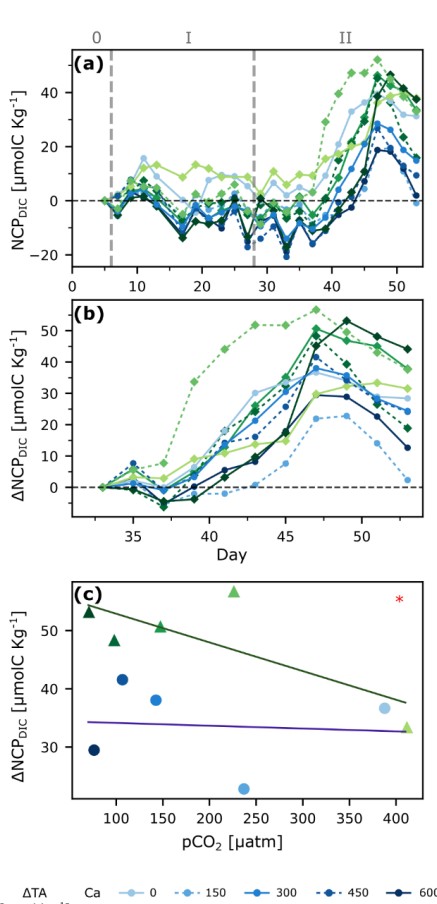

**Figure 4: Primary production under OAE**. Net community production derived from changes in DIC (NCP$_{DIC}$, see 2.6.2) over time (**a**), $\Delta$NCP$_{DIC}$ relative to mean after nutrient addition for a better visualization of bloom peaks (**b**), and regression analysis of bloom peaks against $pCO_2$ levels (**c**). ANCOVA revealed a significant mineral type effect in (**c**): $p<0.05$ (*).

## 4 Discussion

To decouple air-sea gas exchange from biological DIC drawdown and to investigate the effect of different mineral types, we simulated a non-$CO_2$-equilibrated deployment of OAE and followed the development of the system over time. Manipulation of total alkalinity resulted in comparable treatment pairs of Ca and Si treatments with the same $\Delta$TA, which remained relatively stable over time. This is most likely because $\Omega_A$ did not exceed 11.1, the threshold for spontaneous pseudo-homogeneous carbonate formation in mineral particle-free seawater based on





calculations from Marion et al. (2009) at average salinity of 32.6 and 11°C. Further, the TA manipulation led to expected changes in carbonate chemistry speciation, i.e., increased pH and $\Omega_A$, and decreased $pCO_2$, plus no
significant change to DIC (Fig. 1). In the control treatments, DIC declined slightly during the first 10 days, while staying relatively stable thereafter. In contrast to the stable TA, all manipulated treatments increased their DIC content during phase I. DIC drawdown only began in phase II, about 8 days after the nutrient addition. Changes in DIC ranged from about 23 to 49, in contrast to expected Redfield C:N of 20 (6.625*N). Higher values are normally related to carbon overconsumption, which is known to happen in nutrient limited environments and with
increasing temperature (Paul et al., 2016; Taucher et al., 2012). During the bloom in the second phase, pH increases correlate with DIC decreases, suggesting a biological origin, i.e., primary production accompanied by nutrient uptake. Nevertheless, $CO_2$ uptake from air-sea gas exchange needs to be considered as well before drawing further conclusions.

**4.1 Gas exchange**

$F_{CO2}$ started at about -2 mmol $m^2$ $d^{-1}$, indicating that the system was already ingassing $CO_2$ from the atmosphere. This is related to the fact that we arrived at the study site during in a post-bloom period, i.e., seawater $pCO_2$ was lower than atmospheric levels, and the concentrations of dissolved inorganic nutrients was low. After alkalinity addition, daily rates increased between ~ 6 mmol $m^2$ $d^{-1}$ in the lowest treatments and ~-10 mmol $m^2$ $d^{-1}$ in the higher treatments (Fig. 2), which is within the range of air-sea flux estimates from the region (Aalto et al., 2021).
The rates of daily ingassing continued to increase over time, even though the gradient between the atmosphere and seawater $pCO_2$ would gradually decrease with atmospheric $CO_2$ uptake. Since the Schmidt number and the viscosity of gases are influenced by temperature, this is a relevant factor when it comes to diffusion of gases, which in this case may have played a major role. Throughout the experiment, temperature increased from ~8.5 to ~15.5 as we moved from late spring to early summer conditions (Fig. S1). Indeed, there was a statistically signif-
icant positive correlation of temperature and transfer velocity, and a fitting equation was derived to calculate transfer velocities [cm $s^{-1}$] from *in situ* temperature [°C] (Eq. 7). We note that while salinity is also an important factor impacting gas exchange, however, due to the relatively small changes throughout our experiment (~0.1 units), the magnitude of the effect was deemed not-significant for the fitting:

$$k_{CO_2} = -2.67e^{-4} + 9.61e^{-5} * T_{water} \quad (7)$$

In contrast to a mesocosm setup, the most important factor influencing transfer velocity in open ocean settings is wind speed ($\upsilon$). Though it is widely used to parameterize $k_{CO_2}$, it has an uncertainty of ~20% and is valid only for wind speeds in the range of 3 to 15 m $s^{-1}$ (Wanninkhof, 2014). To investigate further, we estimated the wind speed
according to Wanninkhof (1992) and Wanninkhof et al. (2009) using our derived transfer velocities. At our lowest measured temperature, $\upsilon$ was 1.16 m $s^{-1}$, and, at our highest temperature, 1.92 m $s^{-1}$. Furthermore, the mean $\upsilon$ obtained (at mean salinity and temperature) was 1.49 m $s^{-1}$, differing from the mean wind speed near the area of about 3.5 m $s^{-1}$ (Weather Underground: Weather data for IHORDALA29, 2025) This is consistent with the use of the mesocosms, which provide some shelter to the enclosed waters in contrast to the surrounding fjord water,
reflecting that wind speed alone does not drive gas exchange at low wind, but rather alters water surface texture.



In this regard, Eq. 7 would be more suitable than wind speed to estimate $k_{CO_2}$ under similar conditions, i.e., mesocosm setup and similar ranges of temperature and salinity.

Also interesting is that we reached daily rates of $F_{CO_2}$ up to -15 mmol m$^2$ d$^{-1}$ for the highest treatments, which equaled to ~30 µmol C kg$^{-1}$ after a period of 47 days for a volume of 61.6 m$^3$ (20m mixed layer depth). Considering

windspeeds of ~1.5 m s$^{-1}$ and a temperature of ~15.5 °C, we simulated further ingassing until full equilibration (95%) of the mixed layer. We found it would take up to ~1050 days to equilibrate. This timeframe is in line with the findings of He & Tyka (2023), who showed most locations to have an uptake efficiency plateau of 0.6–0.8 molCO$_2$ per mol of alkalinity after 3–4 years.

Therefore, our results indicate that seasons should also be considered, at least in models, for possible deployment

sites, since temperature is an important driver of the exchange process. Thus, in very cold waters there will likely be slower transfer velocities, not forgetting that windspeed and mixed layer depth would also impact equilibration. Lastly, by accounting for the measured air-sea gas exchange, and factoring in the reductions caused by calcification, it is possible to isolate the change in DIC driven exclusively caused by biological activity such as photosynthesis, and respiration.

**4.2 Calcification**

Because of the presence of the coccolithophore *E. huxleyi*, and some availability of inorganic nutrients right at the start of the experiment, there was an initial burst of calcification, identified by a concomitant decrease in salinity-normalized TA, which was then quickly ceased due to nutrient limitation. After the addition of nutrients in phase II, calcification increased again, more pronounced in the lower TA treatments and controls (Fig. S4). When overall

cumulative calcification is related to pCO$_2$ (Fig. 3), we find an optimum curve as predicted in other lab studies for this CO$_2$ range (Gafar et al., 2018). We see then that mild alkalinity treatments (< ΔTA 300 µmol kg$^{-1}$) enhanced calcification, while it was reduced and inhibited at ΔTA 450 and 600 µmol kg$^{-1}$ respectively (pCO$_2$ <100 µatm tested here), this falls within the range of mean sensitivity responses to pCO$_2$ for coccolithophores reported by Seifert et al. (2022). Intermediate pCO$_2$ levels seem to provide an optimal balance of bicarbonate ions and H$^+$

concentration, enhancing calcification rates. Both Krug et al. (2011) and Bach et al. (2011) hypothesized that inhibition of calcification could be the result of substrate (CO$_2$ and HCO$_3^-$) limitation on the one hand, and pH/H$^+$ inhibition on the other. The reason for external H$^+$ constituting an inhibitor is that, during coccolithophoride calcification, H$^+$ is internally being generated (e.g., Gafar et al. (2019); Suffrian et al. (2011); Taylor et al. (2011) and eventually needs to be channeled out of the cell to maintain pH homeostasis (Cyronak et al., 2016). Concern-

ing substrate limitation for both calcification and photosynthesis, it does not matter which carbon species is actually being taken up into the cell as at decreasing seawater pCO$_2$, CO$_2$ leakage out of the cytosol (pH ~7, Anning et al. (1996) will increase due to the concentration gradient. Up to a certain level, this could be compensated for by boosting carbon concentration mechanisms such as active CO$_2$ or HCO$_3^-$ uptake, but this would come at increased metabolic costs (Badger and Price, 2003; Reinfelder, 2011).

Additionally, when cumulative TA-based calcification is compared to cumulative coccolithophore abundance (based on flow-cytometric analysis), there is a linear, statistically significant relationship, supporting the assumption of uniform calcification rates across the coccolithophore population (Fig. S5). There is one treatment (Ca 150) that behaves as an outlier though.



While the cumulative calcification calculated here encompasses overall coccolithophore bloom dynamics, the
response of calcification to changes in carbonate chemistry are typically described by changes in cellular rates.
However, it is possible to link both by calculating the amount of $CaCO_3$ that would be produced in a coccolitho-
phore bloom using rates, derived from lab experiments, termed the $CaCO_3$ production potential (CCPP, Gafar et
al. (2018). Using the rates collected by Gafar and Schulz (2018) for the coccolithophore *E. huxleyi*, which are
dependent on carbonate chemistry, light and temperature, and our average *in-situ* conditions, it is possible to
estimate normalized CCPP and compare it to normalized cumulative calcification (Fig. 3b). Here we see a statis-
tically significant linear correlation, suggesting that the overall bloom dynamics derived from changes in meas-
ured carbonate chemistry align with the behavior at the cellular level derived independently from rates specific to
the coccolithophore *E. huxleyi*. In summary, there is enhanced calcification and CCPP at intermediate levels of
$pCO_2$ (~250 µatm) following a non-$CO_2$-equilibrated addition of alkalinity, which are negatively impacted when
going towards lower and higher levels of $pCO_2$.

**4.3 Net community production as a balance of photosynthesis and respiration**

The *in situ* $NCP_{DIC}$ derived from cumulative changes in the DIC pool (Fig. 4) encompasses autotrophic photosyn-
thesis, decreasing DIC through the consumption of $CO_2/HCO_3^-$, and increasing DIC by both autotrophic and het-
erotrophic respiration. An increase in photosynthetic activity can be observed after the nutrient addition during
phase II, as $NCP_{DIC}$ reached peak values, corroborated by the increase in Chl a (Fig. S1 f). Furthermore, there
appeared to be a general trend of silicon-based treatments reaching higher levels of $NCP_{DIC}$ than the calcium-
based ones (Fig. 4 c). This may be linked to the stoichiometry of inorganic nutrient availability. While in both
treatments' nitrogen uptake was ~3.6 µM, in the silicate treatments, silicate and nitrogen were consumed in a ratio
of up to ~3:1. In contrast, in the calcium treatments, the uptake ratio was reversed at ~1:3, meaning that up to ~10
times more silicate was drawn down in the silicate treatments within 20 days. Since the uptake ratio was 10 times
higher in Si-treatments generally independently of the TA level, it was likely diatoms (if not in quantity, in frustule
thickness) that were heavily silicified, promoting silicification and potential changes to diatom community com-
position, as discussed by Ferderer et al. (2024). Thus, this suggests that diatoms could allocate additional resources
to growth and photosynthesis (Inomura et al., 2023), potentially explaining the higher measured NCP rates (Ma-
rín-Samper et al., 2024) in the silicate treatments and, in turn, contributing to the mineral effect detected here on
$NCP_{DIC}$. In summary, it appears that while there was no negative effect of the TA level on $NCP_{DIC}$, there was a
positive effect of the silicate amendment. This is in line with the hypothesis of the *green* ocean suggested by Bach
et al. (2019), though the *white* ocean enhanced by added calcium was not observed here. One reason could be low
abundances of coccolithophores at the onset of the experiment, the other the fact that increased calcium concen-
trations paired with increased DIC upon equilibration have been hypothesized to promote coccolithophorid calci-
fication and growth (compare Bach, 2015; Bach et al., 2019). However, given the relatively slow air-sea gas
exchange on the order of years and the fact that during this time the TA treated waters are likely subject to sub-
stantial dilution (significantly reducing TA and hence the DIC increase upon full equilibration), a white ocean
might not be something to expect.

The delay in bloom formation observed here (Fig. 4b) was also reflected in *in vitro* oxygen production rates
(Marín-Samper et al., 2024) and was related to both the mineral treatment and the TA level. These delays can be
attributed to the previously reported, species-specific negative relationships between elevated pH/ low $pCO_2$ levels





and phytoplankton growth rates (Chen and Durbin, 1994; Hansen, 2002), as well as the substrate/inhibitor concept already mentioned above. However, in terms of ecological significance, it is not clear if a phytoplankton bloom delay causes knock-on effects for higher trophic levels. Hence, at this stage, it is difficult to draw clear conclusions or provide recommendations.

### 4.4 Zooplankton respiration

The main difference between the estimated *in situ* $NCP_{DIC}$ and NCP obtained from $O_2$ measurements in separate incubations is the exclusion of grazers (larger than 280 µm) in the latter since samples are screened before incubation. So, when comparing both, it is possible to get an estimation of the contribution of larger zooplankton and fish respiration to the carbon balance (Eq. 8):

$$RZ = NCP - NCP_{DIC} \; , (8)$$

with RZ denoting zooplankton respiration and NCP the cumulative net community production derived from $O_2$ incubation measurements, assuming a 1:1 $O_2$:C conversion ratio. Cumulative RZ ranged from 30-60 µmol kg$^{-1}$ and was significantly correlated to $pCO_2$ as well as to mineral treatment (Fig. 5). Interestingly, the treatments with lower $NCP_{DIC}$ had higher RZ, possibly indicating a top-down control on primary productivity. Reasons for this observation could be direct effects of high pH at low $pCO_2$ on zooplankton metabolism (Hessen and Nilssen, 1983; Pedersen and Hansen, 2003), if requiring more energy along the TA gradient to maintain homeostasis under suboptimal conditions, or indicating effects of changes to phytoplankton community composition, influencing prey availability between the two mineral types. Furthermore, under low $pCO_2$ conditions, phytoplankton have lower C:N content (Burkhardt et al., 1999), which increases their quality as food for zooplankton, likely leading to higher grazing and respiration rates. All this suggests that the response to carbonate chemistry perturbations might also be species-specific and that a top-down control on primary productivity might have been exerted. Interestingly, Goldenberg et al. (2024) showed that though fish abundance in our experiment did not correlate to the treatment gradient, biomass did. Although no clear explanation could be found by the authors, it highlights the complex trophic interactions in pelagic ecosystems that require untangling. Finally, a recent study showed enhanced copepod grazing rates for certain OAE scenarios, although respiration rates didn't seem to be correlated (Bhaumik et al., 2025), again highlighting the lack of mechanistic understanding of the underlying processes.





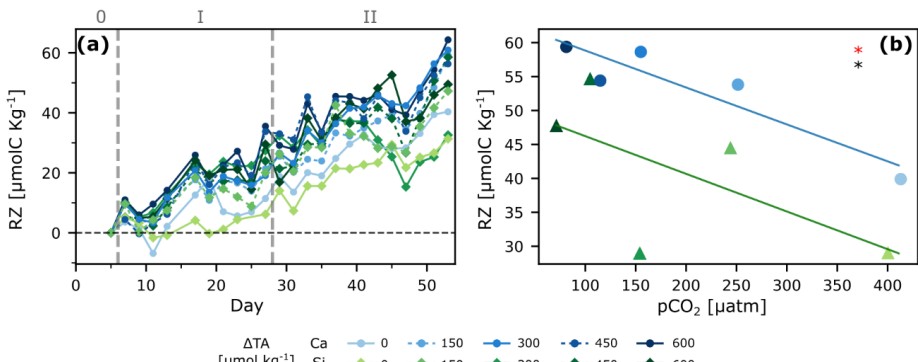

**Figure 5: Zooplankton under OAE.** Respiration of Zooplankton (RZ) over time (**a**), and regression analysis of the last 2 days (averaged) vs $pCO_2$ (**b**). ANCOVA results: OAE (treatment) effect: $p < 0.05$(*), and mineral type effect: $p < 0.05$(*).

**Table 1: Summary of responses under non-$CO_2$-equilibrated OAE. "="** indicates no effect detected, while "✓" indicates a detected effect. The arrows depict the direction of the response to the TA gradient (decreased, ↓; enhanced, ↑).

| Response variable | Mineral type effect | OAE effect | Remarks |
|---|---|---|---|
| $CO_2$ ingassing | = | ↑ | Increased with decreasing $pCO_2$<br>Up to 15 mmolC m$^{-2}$ d$^{-1}$ |
| Coccolithophorid Calcification | = | *↓ | *Optimum curve response, peaking at ΔTA 150 µmol kg$^{-1}$ and $pCO_2$ ~250 µatm, then decreasing |
| Net Community Production | ✓ | = | More pronounced in Si treatments |
| Zooplankton Respiration | ✓ | ↑ | Higher in Ca treatments<br>Increased with lower $pCO_2$ |

**5 Conclusions and outlook**

Our study shows that $CO_2$ ingassing can occur at rates of up to 15 mmol C m$^{-2}$ per day after a non-$CO_2$-equilibrated deployment of OAE. This equates to ~30 µmol C kg$^{-1}$ over 47 days for a 61.6 m³ volume under low wind conditions, reaching full equilibration after ~1050 days. Furthermore, phytoplankton responses showed an optimum curve for coccolithophorid calcification with peaks at mild treatments (TA 150 µmol kg$^{-1}$, $pCO_2$ ~250 µatm), aligning with previous laboratory predictions. No significant effect on NCP$_{DIC}$ was observed with lowered $pCO_2$, but a mineral effect was noted, with maximum NCP$_{DIC}$ being more pronounced in Si treatments, potentially due to enhanced Si(OH)$_4$ concentrations. Lastly, zooplankton respiration was lower in silicate treatments and increased with lower $pCO_2$, indicating species-specific responses and potential top-down control on primary productivity. Based on our findings (Table 1), we conclude that under a scenario without coinciding silicate addition, an OAE






application would be unlikely to have a significant impact on the plankton community up to levels around $\Delta$TA of 150 $\mu$mol kg$^{-1}$. At higher TA levels, in particular in conjunction with added silicate, there might be complex interactions among multiple trophic levels, requiring further disentangling.

**Data availability**

All data supporting this article will be available for open access on the PANGAEA data portal upon publication.

**Supplement link**

(to be added by Copernicus)

Figure S1: Temporal development of environmental conditions

Figure S2: pH correction/calibration

Figure S3: $N_2O$ measured concentrations and chemical enhancement

Figure S4: Net community production related figures

Figure S5: Calcification related figures

**Author contribution statement**

UR, KGS and LRK designed and conceptualized the mesocosm experiment. JS, KGS, CAM, LMS, LRK and JOC collected and analyzed samples in the laboratory. JS and KGS were responsible for data curation and formal

analysis. JS, KGS, CAM and LMS interpreted results. JS prepared the original draft with particular input from KGS and contributions from all co-authors.

**Competing interests**

The research reported in the manuscript was conducted during academic activities, prior to the start of other employment. Julieta Schneider has been consulting for the start-up Planeteers GmbH, Germany, as a Geochemical

Researcher since November 2024, and Joaquín Ortíz-Cortes is employed by Macrocarbon S.L., Spain, since October 2023.

**Acknowledgements**

We would like to thank the University of Bergen, Marine Biological Station Espegrend, for the use of their facilities and help with logistics. This study involved huge team effort, so we are grateful to the staff and students from

the KOSMOS team (GEOMAR) and all study participants for their contributions on site. In particular we thank: Andrea Ludwig and Jana Meyer for logistical support and coordination of on-site activities; Anton Theileis and Jan Hennke for mesocosm preparation, technical support and maintenance; Daniel Brüggemann, Philipp Süßle, Joaquin Ortiz, Nicolás Sánchez, Carsten Spisla and Michael Sswat for onsite scientific diving activities and maintenance. Extended thanks go to Juliane Tammen and Peter Fritzsche for the measurement of dissolved inor-

ganic nutrients, to Nwafor Chukwudi for measurement of the $N_2O$ samples in Kiel, and to Niels Suitner for the interesting discussions on data interpretation.



**Financial support**

This study was funded by the OceanNETS project ("Ocean-based Negative Emissions Technologies – analyzing the feasibility, risks and co-benefits of ocean-based negative emission technologies for stabilizing the climate", EU Horizon 2020 Research and Innovation Program Grant Agreement No.: 869357), and the Helmholtz European Partnering project Ocean-CDR ("Ocean-based carbon dioxide removal strategies", Project No.: PIE-0021). Huge support from the AQUACOSM-plus project (EU H2020-INFRAIA Project No.: 871081, "AQUACOSM-plus: Network of Leading European AQUAtic MesoCOSM Facilities Connecting Rivers, Lakes, Estuaries and Oceans in Europe and beyond") was provided as well.





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
