# Peer review of "Air-sea gas exchange measurements helped derive in-situ organic and inorganic carbon fixation in response to Ocean Alkalinity Enhancement in a temperate plankton community"

_EGUsphere, 2025_

## Referee Comment (RC1)

**Review of Schneider et al. (2025): "Air-sea gas exchange measurements helped derive in-situ organic and inorganic carbon fixation in response to Ocean Alkalinity Enhancement in a temperate plankton community"**

The paper by Schneider et al. investigates the ecological and chemical consequences of non-$CO_2$-equilibrated Ocean Alkalinity Enhancement (OAE) using a mesocosm experiment in a temperate fjord. The study's overarching hypothesis is grounded in existing literature, particularly the "white vs. green ocean" framework (Bach et al., 2019), and addresses known gaps in empirical OAE studies. Specifically, it aims to test whether varying levels of added total alkalinity (TA) using silicate- and calcium-based minerals alter carbonate chemistry, air-sea $CO_2$ fluxes, calcification, net community production (NCP), and zooplankton respiration. This hypothesis is both relevant and timely, as most OAE research remains conceptual or model-based. Some explicit strengths, use of Gafar et al.'s (2018) $CaCO_3$ Production Potential (CCPP) bridges cellular physiology with mesocosm-scale data. Estimation of zooplankton respiration as the difference between oxygen-based and DIC-based NCP is innovative and revealing.

**2. Experimental Design and Methodological Soundness**

The methodology needs to be improved and supplemented with the method description (and results) currently missing.

- One of the biggest limitations is that there was only one direct DIC sampling point for validation introduces uncertainty into TA-pH derived DIC estimates. Such uncertainties have likely propagated through time but that has not been addressed in the paper. In addition, NCP is derived from changes in DIC — so uncertainty compounds over time. This needs to be addressed and evaluated.

- Methodology of how preparing different feedstock solutions is not described in sufficient details. Artificial separation of silicate and TA effects may not represent real-world OAE mineral additions. How were the concentrations of all the feedstocks measured to assure that the concentrations at the end were correct?

- How much $NO_3^-$, Si and $Si(OH)_4$ and $Ca^{2+}$ were added, mention specific numbers. Why was $NO_3^-$ added to up to 4umol/kg, which is at least 4-5 times higher than in a fjord, creating completely artificial conditions for those communities inhabiting fjord? And subsequently, how do you know that this is a natural response of the communities acclimatized to low nutrient levels, instead of artificial response that might be out of scope if OAE without the added nutrients would happen? Can you decouple this effect somehow and include this in the discussion and results section?

- What depth was $N_2O$ taken, up to 20 m or the surface, not clear from the text.

- Respiration measurement description is missing.

- The description or the reference to the flow cytometry analyses is missing.

**3. Data Collection and Analytical Approach**

**Analytical concerns:**

- The pH measurements required dye corrections due to potential impurity artifacts— highlighting the fragility of spectrophotometric pH at high alkalinity. Can you comment and revise?

- Assumption of 1:1 $O_2$:C ratio in NCP calculation may oversimplify complex respiration dynamics. How do you rectify this? In which range does this ratio hold? Could that be different for the respiration of the micro vs large zooplankton (above 280um)?

- Data on respiration is missing entirely.

- The large variability of DIC upon the nutrient addition is overwhelming and not well explained, also not matching the trends in the other parameters. Provide better explanation.

- Where did you take the 95% for full equilibration from?

- No coccolithophores or diatom data presented??? It is literally impossible to draw some of the results and conclusion in this paper unless there is data available for this.

- Are there any taxonomic or metagenomic assessments to resolve zooplankton community and why the decision on cutting it at 280um?

**3. Results and Interpretation**

**Calcification:** Coccolithophore calcification followed an optimum curve relative to pCO2, with a peak around 250 µatm and suppression at extremes. However, in the figure S4a, calcification is below 0 for the two highest treatments, which is not explained anywhere in the text. Does this indicate dissolution. Even less severe treatments are still just hardly above 0, especially before the addition of nutrient part, which signifies lack of calcification overall, and only just happening in the first three treatments. How does this align with the NCP, can you correlate? And how does it align with the CALC, could it have any effect on the TA? Is this species-specific, could it be due to any other calcifiers (not just the autotrophs)? In general, the drawback of this is also that no other potential calcifiers have been implicated in the CALC, only the autotrophs. Are there any data available to support this, or discount for the impact of zooplankton on the CALC?

**Net Community Production (NCP):** NCP was significantly higher in silicate treatments post-fertilization, with no direct pCO2 effect. But the effect of the pH was not investigated and should be included in ANOVA. Also, why is NCP related to Si and not to $Ca^{2+}$ treatment- again, data on diatoms and phytoplankton are absolutely essential, otherwise this all on the level of inferences.

Also, how does Chla correlate with NCP and Calcification (Figure S4a-c and S1f)?

**Zooplankton Respiration:** Respiration declined with decreasing pCO2 and was lower in Si treatments, but in general, this aspect is largely underexplored and insufficiently presented. Much more effort needs to be put in explaining respiration data and how it links to suggested trophic-level complexity. Present the data on respiration beyond 2 days, 2-day data is insufficient, compare the pre and post nutrient treatment.

The results of respiration are also fundamental in explaining some of the effects and should be put in the Results, not Discussion, section.

**Discussion:**

In general, this study is really divided in two parts:

- Pre-nutrient treatment that is represented of the fjord environment under OAE and post-nutrient that is NO LONGER representative of the oligotrophic fjord conditions, whereby the used communities were not acclimated to such increases in nutrients and is just a mesocosm trial of OAE with nutrients. In such systems, the communities and species could react completely differently than under such artificial conditions. This aspect is now touched upon in the results and discussion and I would like the authors to fully dedicate the effort on the potential confounding effects due to such nutrient addition and how different the fjord system response to OAE would be if such strong nutrient artificial addition was not present- Fco2 still high, but NPC insignificant, what about respiration etc?

- In addition, no evaluation of the longer-term dynamics to capture seasonal or successional effects is presented.

---

## Referee Comment (RC2)

Review:

**Schneider et al.: „Air-sea gas exchange measurements helped derive in-situ organic and inorganic carbon fixation in response to Ocean Alkalinity Enhancement in a temperate plankton community "**

**Key results**

The authors investigated the effects of ocean alkalinity enhancement (OAE) on pelagic communities in a temperate fjord. In particular, they investigate carbon dioxide removal efficiencies as well as community responses to gradients of two different alkaline minerals (calcium- and silicon-based). Their results show that the amount of added alkalinity modifies the CO2 removal capacity, independent of the mineral. Furthermore, calcification is enhanced under medium alkalinity additions and net community production is stimulated by the addition of the silicon-based mineral, likely due to increasing diatom productivity.

The study is timely and important to improve the understanding of ecosystem effects of OAE, especially as knowledge on environmental impacts of OAE is still very limited. The authors made big efforts to analyse and understand the dynamics in their mesocosms, using a variety of methods. The investigation of the carbon removal efficiency as well as ecosystem impact of two different alkaline materials, inspired by the white vs. green ocean hypothesis of Bach et al. (2019), is a vital contribution to move OAE research closer to actual application. The findings of the study can be an inspiration for similar studies in other ocean regions, as well as for more detailed investigations of the separate processes.

Although I strongly encourage publication, I recommend major revisions prior to acceptance. In the following I add my main issues for each of the sections. Below you can find line-by-line comments. My main comments concern the presentation and discussion of the results and not the experimental setup in general. I have to add that I'm not working in the field or laboratory myself and cannot fully evaluate the methodologies in this study.

**General comments**

The title is very descriptive. It's a matter of taste, but I would recommend to include your main findings, e.g. the optimum functional response of coccolithophores to alkalinity addition.

The abstract is concise, although the description of the results could in parts be improved, see line-by-line comments.

The introduction successfully introduces the reader into the functioning of ocean alkalinity enhancement. The authors could improve the last part by removing some of the method description and adding more details on the motivation to conduct experiments with different alkaline material under nutrient addition.

The results could be restructured to highlight similarities and differences in the carbonate system, calcification rates, NCP etc. of the different experiments (i.e., Si vs Ca-based mineral and the effects of nutrient addition), sticking to the initial research questions of the study. In parts this is already done, but somewhat hidden in additional calculations and assumptions which could rather be moved to the result (and partly discussion) section. A clear structure and a consistent naming of the experiments would help the reader to understand the basic results of the study. I do like Table 1, which nicely summarizes the findings of the study.

The analysis of the carbonate system changes reads a bit like all over the place. I suggest to carefully disentangle the different measurements you have (TA, DIC, pCO2, FCO2) and what you can learn from these changes (e.g., are they biologically mediated or caused by physics and resulting changes in the carbonate system).

I couldn't find a comparison of CALC and CCPP as announced in L. 246-247 (apart from Fig. 3b) and an interpretation of this. I'm not sure if this is relevant for the manuscript, but it should at least be added to the supplement.

The discussion is very close to the results of the study, which is good, but could me moved from a description of the results to an interpretation of overall findings embedded into existing literature. In parts this is done, but it is hidden in a repetition of the result.

Conclusion and outlook are rather a summary of Table 1 than a real perspective on the study. I suggest to think about what we can learn from the findings of the study, what could be potential follow-up experiments, and which advices you would give for large-scale applications of OAE.

Finally, not being a native English speaker myself, I would recommend to do a thorough language check as there are some grammar and phrasing issues here and there.

**Line-by-line comments**

Abstract

L. 15-16: Reduced pCO2 and pH only until equilibrium is reached, right? Maybe it's worth to add this to the sentence.

L. 17: Better: "in *a* temperate fjord"

L. 23: "with cumulative coccolithophore calcification showing an optimum curve response to decreasing pCO2" – this sounds very complex. Can you simplify the message?

L. 24: "while no mineral-type effect was observed" – compare to first part of the sentence ("there were a number of mineral-type and pCO2/pH effects"); this seems to be contradicting. Perhaps splitting the sentence in two parts would help?

L27-28: "These complex findings suggest both direct and indirect effects of mineral type and OAE level and provide a valuable foundation for designing future OAE field trials" – can you be more specific about the message of your findings? For example, carbonate-minerals should be favored over silicate-minerals (smaller environmental impact) and the amount of added alkalinity must balance effectivity ($CO_2$ flux into the ocean) and environmental impact (calcification).

**Introduction**

L. 44-46: "This variability…" – yes, but the variability that you mention in the previous sentence only refers to modelling studies. There are a number of reasons why modelling studies vary in their estimates of $CO_2$ uptake potential, in the end they assume different alkalinization scenarios etc. I agree that the $CO_2$ uptake potential of real-ocean applications is difficult to predict because we do not know enough, but then you need to rephrase the sentence.

L. 50: "Since gas exchange *can* take …"

L. 60: "all of which release soluble products of interest for marine organisms" – unclear; why are the alkaline minerals "of interest" for marine organisms?

L. 61-64: "For these materials …" – this sentence is very long and quite confusing, can you rephrase? For example, it is currently unclear what you mean with white and green ocean for readers who don't know the study of Bach et al. You could also emphasize more that you aim to test the white vs green ocean hypothesis of Bach et al., this is currently somewhat hidden in your statements.

L. 67ff: I suggest to shorten, parts of this are methods.

L. 74: "These perturbations would be highest close to an OAE point source" – okay, but you don't test OAE effects with distance to a point source in your study, right? I suggest to shorten here.

L. 77-78: "Processes that need to be better understood before large field deployments of OAE and for later monitoring and verification." – Please check, this is not a proper sentence. And here you could add a few more details – which processes do you aim to understand? What is the goal of your studies, which parts of the ecosystem do you assess?

**Material and Methods**

L. 88: "surrounding" (without *s*)

L. 91: "to remove any heterogeneously distributed nekton" – what is this? Wouldn't "remove nekton" be enough?

L. 92-94: "Additionally, a volume determination …" – this sentence is unclear; do you measure the volume change or the total volume, and why is this important? Please rephrase.

L. 99-101: As a reader who is not familiar with OAE experiments I would wonder why you don't just add lime and olivine power?

L. 103: "*in* the olivine scenario"

L. 109-111: "(the unavoidable …)" – this sentence is unclear. Can you state more clearly that the addition of silicate leads to a shift in alkalinity that is not wanted in your experiments because you want to control TA only by the addition of NaOH? At least this is how I understand your description, but it took a while until I got your message.

L. 119-121: Why is the concentration of nutrients lower than in the surrounding water? And what is your motivation to add nutrients? This has not been explained in the introduction, has it? Furthermore, please check the sentence "Nitrate concentrations …" – is it possible to describe the amount of added nitrate and phosphate in a similar manner?

L. 122: "Ca treatments" – I recommend to use uniform terms for your experiments throughout the manuscript.

L. 123-124: "Nutrient concentrations …" – please check sentence structure and grammar.

L. 128: What exactly do you mean with "randomized sampling"?

L. 149-150: change "The filtration process resulted in the removal of biomass" to "The filtration process aimed at removing" or "The filtration process removed"

L. 151f: "Water for gas-sensitive parameters…" – please check the sentence structure.

L. 161: To compute average and standard deviation you would need triplicates, not duplicates. Is there a reason why you did not perform three measurements?

L. 171: Change "Said samples" to "Samples"

L. 196: "Throughout the experiment, …" – please check the wording. "Coupled" does not sound correct in this context.

L. 202: Please check the section title. The part "and community production estimates" appears also in the next section title in line 234, and this is where you actually describe community production.

L. 242: It would be helpful if you could give a unit for CALC.

L. 244: How do you know that coccolithophores are the main calcifiers in your mesocosms? I suppose that your experiment was performed jointly with other experiments in which the community composition was analysed, but if so it should be mentioned.

L. 244-245: Can you give an equation and a unit for CCPP? Which light conditions do you mean? Surface or at a specific depth?

L. 249: Maybe I missed this, but where do you have cCO2 from?

L. 254: "The following change in sign …" – I know what you mean, but the phrasing is a bit odd. Consider to rephrase to something like "We define NCP to be positive" or alike.

L. 255: The description of NCP from O2 measurements is missing.

L. 260: "for 5 additions of increasing dye addition" – rephrase to "for 5 additions of increasing dye concentration"

Results

L. 272: "After the manipulation on day 7" – but the manipulation took place on day 6? It would be more precise to say "After the manipulation on day 6" on "On day 7, one day after the alkalinity manipulation".

L. 273: "while pCO2 decreased to 61" – unit is missing.

L. 273: "In contrast, DIC was hardly affected." – Compare to Fig. 1d and L. 277: DIC is considerably affected by the alkalinity manipulation (and nutrient addition).

L. 274: "Overall, TA remained relatively stable throughout the experiment, regardless of the phase and the treatment" – this is only true for phase I and II, but not between phase 0 and I.

L. 276: "from day 35 onwards" – it would be easier for the reader if you would consistently refer to the phase and/or the respective manipulation (here: "after the nutrient addition in phase II").

L. 280: "DIC decreased in all mesocosms again, …, coinciding with average Chl a changes" – I suggest to mention that DIC is decreasing as soon as Chl a increases (i.e., different direction of change).

L. 289: "All treatments showed an immediate uptake of CO2" – where do you see this? In FCO2?

L. 290: "While FCO2 didn't change significantly for the control treatment right after the TA addition" – but you did not add TA to the control treatments, did you? TA should not change in these treatments anyway? Generally, I feel that you could use the control treatments a

bit more as such, i.e., you could analyse the changes in the other treatment relative to the control treatment to make sure that the only difference is the addition of the alkaline material and no other changes in the system that may play a role in the course of the experiment (being aware that experimental treatments are not 100% replicates of each other).

L. 290-291: "it increased with increasing deltaTA and corresponding lower seawater pCO2" – FCO2 is decreasing, right? Double-check sign convention. I also recommend to rephrase this sentence as it currently implies that deltaTA in the control treatment is increasing, but you mean the different TA-addition treatments.

L. 294: "Notably, no differences in FCO2 were observed between the different minerals" – This is not entirely true, I do see differences in Fig. 2. They may not be statistically significant, but you should not say "no differences".

L. 299: "Except for the Ca-treatment deltaTA 150" – I like the idea of giving simple but specific names to the different treatments, but they should be introduced and consistently used throughout the manuscript. Indeed, this could simplify some phrases in the remaining manuscript.

L. 303: How about the comparison of CALC and CCPP as displayed in Fig. 3b?

L. 304: "at the beginning" – what exactly do you mean? Phase 0?

L. 308: Are you sure that you show cumulative changes? What is negative cumulative NCP then? More respiration than production?

L. 310-311: "Though such a statement assumes that …" – important to mention, but I suggest say a few more words on this and perhaps move to the discussion, possibly even to a a separate "limitation" section.

L. 314: "An ANOVA analysis showed no alkalinity effect, but a clear mineral effect on production" – if you tested NCP vs. pCO2 as described in the previous sentence, it would rather be a pCO2 effect and not an alkalinity effect.

Fig. 4c: Why triangles here? They should be diamonds, right?

Discussion

L. 333ff: Because it is an integral part of your experimental setup (although I miss a clear motivation for this as mentioned earlier) I suggest to add the nutrient addition to the sentence.

L. 335-336: "Manipulation of … which remained relatively stable over time" – please specify, what remained stable over time? DeltaTA? The treatment pairs?

L. 336: Hm yes, but as you mention correctly an omega aragonite of 11.1 may be the threshold under idealized conditions. However, the ocean is not particle-free (plus you have different temperature and salinity values compared to the idealized study of Marion et al. 2009), and these particles can serve as precipitation nuclei, decreasing the threshold of abiotic CaCO3 precipitation. It seems that this is not the case in your experiments, but you could add this argument in your discussion.

L. 339: "plus no significant change to DIC" – this is confusing, you do see quite some changes in DIC?

L. 341: "all manipulated treatments increased their DIC content" – I suggest to rephrase, the treatments did not actively increase the DIC content.

L. 343: "Changes in DIC ranged from about 23 to 49, in contrast to expected Redfield C:N of 20" – I cannot follow this argumentation, which DIC change would you have expected?

L. 351: "This is related to the fact that we arrived at the study site during in a post-bloom period" – it doesn't really matter when you arrived ;-) Perhaps rephrase to "the experiment was started during a post-bloom period" (also note that you have a phrasing typo: "*during in a post-bloom...*")

L. 353: "daily rates increased" – note sign convention, currently it would be "decreased".

L. 354: "which is within the range of air-sea flux estimates from the region" – what do you mean, is the additional CO2 uptake after alkalinity addition not caused by ocean alkalinization but is rather caused by natural variations of the CO2 uptake?

L. 358-359: Temperature unit is missing (I guess it is °C).

L. 381-382: "In this regard, …" – this statement is unclear to me. Increasing ocean CO2 uptake is more likely explained by changes in temperature than by changes in wind speed over the time of the experiment because estimated wind speeds do not fit to observations?

L. 384: "Therefore, our results indicate that seasons should also be considered …" – but if the equilibration time takes more than 1000 years, passing through all seasons of almost 3 years, does the day of deployment really matter? Furthermore, I do not agree with the statement "at least in models", because models should try to simulate what could be done in the real ocean.

L. 387-389: "Lastly, by accounting for the measured …" – this is what you will do in the following, right? If yes, you can add this to the sentence.

L. 391: "Because of the presence of the coccolithophore E. huxleyi …" – how do you know that Ehux is growing in your mesocosms? I assume that a community analysis was done in the framework of other studies on the same mesocosms, but this should be mentioned somewhere (here or in the methods). Otherwise, the certainty that Ehux is the main calcifier

in your experiments comes out of nowhere. You could, for example, give a short summary of the community analysis including relative contributions of all planktonic calcifiers.

L. 403 and L. 407: Right parenthesis missing

L. 390: Do you observe this bloom only in the experiment with alkalinity manipulation or also in the control experiment? I suggest to be more specific here, because it can be a bit confusing to understand which changes are caused by the alkalinity / nutrient manipulations and which changes are part of the natural change in the fjord.

L. 410: "Additionally, when cumulative TA-based calcification is compared to cumulative coccolithophore abundance …" – what you want to say is that the per cell calcification remains constant but the number of cells is increasing? Hence, the optimal conditions for calcification rather lead to proliferation than to heavily calcifying cells? This is an interesting finding and could be highlighted more. Although I got a bit confused by the following discussion on cellular rates.

L. 433: typo in "treatments"

L. 433: "nitrogen uptake was ~3.6 µM" – cumulative? Per unit time?

L.436-437: "… it was diatoms (if not in quantity, in frustule thickness) that was heavily silicified …" – in the second part of this sentence you imply that diatoms increase their frustule thickness and kind of exclude that the strong Si uptake can be caused by a proliferation of diatoms. If you don't know for sure I suggest to keep this open. But maybe you do have an idea as it seems like you have an impression of the community composition, given your previous statements on Ehux calcification?

L. 438: "Thus, this suggests that diatoms could allocate additional resources to growth and photosynthesis …" – doesn't this just show that diatoms were Si-limited previously?

L. 440: "… in turn, contributing to the mineral effect …" – is "contributing to" the correct word here? Shouldn't it rather be "causing"?

L. 444: "One reason could be low abundance of coccolithophores at the onset of the experiment …" – okay, but you do see an increase in coccolithophore abundance in the experiment under medium alkalinity additions (see for example L. 423-425)? The remaining sentence ("the other the fact that ….") is unclear to me, please rephrase.

L. 450: "The delay in bloom formation observed here" – this is a bit unspecific and as a reader I have to go back to the results to remember what you mean by "here". It would be easier if you would recap that you are talking about the delayed bloom peak in the high-alkalinity experiments. In the same sentence, you refer to the study of Martín-Samper et al. 2024. Is that study using the same experimental setup or even the same mesocosms? Or is that study completely independent from yours? This is currently unclear.

L. 450-456: This entire paragraph needs some more details. Why do you think the delay of the bloom could be caused by the substrate/inhibitor concept? What is a "knock-on effect"? What are the open questions and which additional considerations are required to be able to provide recommendations?

L. 458: As mentioned above, "NCP obtained from O2 measurements" was not described in the methods.

L. 458-466: Consider to move to the results and/or methods.

L. 467-468: "Interestingly, the treatments with lower NCPDIC had higher RZ, possibly indicating a top-down control on primary production." – you can make it easier for the reader if you repeat which experiment resulted in lower NCPDIC (or higher NCP?). Also, you will always have a top-down control on primary production in a natural system, won't you? Or do you want to say that you observe a "stronger" top-down control / more grazers in the system?

L. 468-472: "Reasons for this observation could be …" – here you mix the arguments for a weaker phytoplankton-zooplankton link due to increasing alkalinity addition and the effect on zooplankton metabolism (first part of the sentence) and due to mineral-mediated changes in the phytoplankton community composition (second part of the sentence), do you?

L. 475: But you do not investigate single species? How can you draw the conclusion that responses are species-specific? It is probably true, but I think the jump from your results to this conclusion is a bit too big.

L. 475: " … and that a top-down control on primary productivity might have been exerted" – this sentence is unclear to me.

L. 475: Here again, how comes that Goldberg et al. 2024 knew what is going on in your experiments? If this study is based on the same mesocosm experiments than yours, it should be mentioned. Furthermore, the link between your findings and the findings on fish biomass vs abundance is not completely clear to me.

L. 479: "…. although respiration rates didn't seem to be correlated (Bhaumik et al. 2025)." – add: "to be correlated to the amount of alkalinity added" or similar.

Table 1: A very nice summary table!

Conclusion and outlook

L. 496: "potentially due to enhanced Si(OH)4 concentrations." – consider to add "and the concomitant proliferation of diatoms"

L. 498: "indicating species-specific responses and potential top-down control on primary production" – see comments above.

---

## Author Comment (AC2)

**Response to comments from Reviewer 1**

We appreciate the reviewer's positive feedback and thank them for their time and effort. We have addressed each of the comments below, with our responses introduced in italics and labelled as *Authors' Response (AR)*.

**General Comments:**

The paper by Schneider et al. investigates the ecological and chemical consequences of non-CO2-equilibrated Ocean Alkalinity Enhancement (OAE) using a mesocosm experiment in a temperate fjord. The study's overarching hypothesis is grounded in existing literature, particularly the "white vs. green ocean" framework (Bach et al., 2019), and addresses known gaps in empirical OAE studies. Specifically, it aims to test whether varying levels of added total alkalinity (TA) using silicate- and calcium-based minerals alter carbonate chemistry, air-sea CO2 fluxes, calcification, net community production (NCP), and zooplankton respiration. This hypothesis is both relevant and timely, as most OAE research remains conceptual or model-based. Some explicit strengths, use of Gafar et al.'s (2018) CaCO3 Production Potential (CCPP) bridges cellular physiology with mesocosm-scale data. Estimation of zooplankton respiration as the difference between oxygen-based and DIC-based NCP is innovative and revealing.

**Specific Comments:**

**2. Experimental Design and Methodological Soundness**
The methodology needs to be improved and supplemented with the method description (and results) currently missing.
- One of the biggest limitations is that there was only one direct DIC sampling point for validation introduces uncertainty into TA-pH derived DIC estimates. Such uncertainties have likely propagated through time but that has not been addressed in the paper. In addition, NCP is derived from changes in DIC — so uncertainty compounds over time. This needs to be addressed and evaluated.

*AR: We agree that it is a limitation of the study to only have 10 direct DIC measurements, from which a correction of measured pH across the gradient could be obtained, which then was used throughout the study to calculate DIC. However, the correction at the highest pH level of 8.5 is 0.03 pH units (see Fig. S2 in original supplement) which, in turn, translates to a calculated DIC offset of about 25 µmol kg$^{-1}$ (only ~1.2%). The fact that DIC remained relatively stable in the days following the TA addition—consistent with expectations for this oligotrophic phase of the experiment—gives us confidence that the DIC estimates based on corrected pH measurements are reliable. Furthermore, even if there were a tendency for increasing offsets in calculated pH in the higher TA treatments, this would hardly affect our NCP calculations, as they are based on relative change over time. Finally, the effects observed in NCP are only mineral-type, which would not change if all values are skewed. We will include these clarifications in a separate paragraph in the discussion.*

- Methodology of how preparing different feedstock solutions is not described in sufficient details. Artificial separation of silicate and TA effects may not represent real-world OAE mineral

additions. How were the concentrations of all the feedstocks measured to assure that the concentrations at the end were correct?

*AR: The preparation of the feedstock solutions is laid out in lines 106-111. We will add further information, such as that the Ca and Mg solutions, as well as the NaOH were all prepared in individual 20 L of deionized water and then added to the respective mesocosms. Mg and Ca were not measured, as deemed a rather small change of only a few percent, given the large natural background concentrations (~49.8 and 9.6 mmol kg$^{-1}$at salinity 33, respectively). However, the increase in silicate and TA by NaOH was confirmed by direct measurements in the mesocosms right after the additions. Concerning the fact that artificial separation of silicate and TA may not represent real-world mineral additions, even at the lowest TA addition of 150 µmol kg$^{-1}$, the silicate addition for a correct olivine stoichiometry would have resulted in an increase of silicate by 37.5 µmol kg$^{-1}$. This is more than an order of magnitude larger than what is considered to be limiting for diatom growth. Furthermore, only about 10 mol kg$^{-1}$ of silicate was taken up until the end of the experiment, meaning that in any case, silicate concentrations would have been non-limiting throughout the experiment in all mesocosms. We will add this additional information to the methods section.*

- How much NO3-, Si and Si(OH)4 and Ca2+ were added, mention specific numbers. Why was NO3- added to up to 4umol/kg, which is at least 4-5 times higher than in a fjord, creating completely artificial conditions for those communities inhabiting fjord? And subsequently, how do you know that this is a natural response of the communities acclimatized to low nutrient levels, instead of artificial response that might be out of scope if OAE without the added nutrients would happen? Can you decouple this effect somehow and include this in the discussion and results section?

*AR: We will add the information that Ca and Mg were added in a 1:2 ratio to TA. Concerning the nutrient additions, we will refer to Ferderer et al., 2024 for further and specific details. When it comes to upwelling events bringing nutrients to the surface, these are not uncommon in the study area and similar nutrient additions have been carried out in a number of past mesocosm studies there, e.g., Schulz et al. 2008, Schulz et al. 2017. The latter study has also highlighted that nutrient additions halfway through an experiment tend to amplify otherwise difficult to detect differences in community composition/biogeochemical element cycling. We will add this information to the discussion.*

- What depth was N2O taken, up to 20 m or the surface, not clear from the text.

*AR: N$_2$O sub-samples were taken from the IWS (Integrating water samplers), which integrated water from 0-20 m depth. We consider this is sufficiently clear between lines 129 and 132 (section 2.4).*

- Respiration measurement description is missing.

*AR: If the referee refers to Zooplankton respiration, we consider this to fit better in the discussion section, as it came only as a result of comparing our DIC-derived NCP with oxygen-derived NCP from a different publication. No measurements were performed.*

- The description or the reference to the flow cytometry analyses is missing.

*AR: We will add a brief description of the Flowcytometry analysis.*

**3. Data Collection and Analytical Approach**
**Analytical concerns:**

- The pH measurements required dye corrections due to potential impurity artifacts—highlighting the fragility of spectrophotometric pH at high alkalinity. Can you comment and revise?

*AR: We wouldn't say that spectrophotometric pH is fragile, since the change in absolute values was only 0.03 pH units at the highest measured pH of 8.5 (see response above). Such pH dependent offsets using unpurified dyes, even when trying to apply corrections have been described previously (e.g., Douglas & Byrne 2017). It highlights the advantage to over-determine the carbonate system by measuring 3 rather than only 2 parameters for cross-validation. We will add the reference to the appropriate methods section.*

- Assumption of 1:1 O2:C ratio in NCP calculation may oversimplify complex respiration dynamics. How do you rectify this? In which range does this ratio hold? Could that be different for the respiration of the micro vs large zooplankton (above 280um)?

*AR: Indeed, the reviewer is correct that the trends we see in calculated zooplankton respiration could also be the result of changes in the respiratory, as well as the photosynthetic quotient. We will discuss this in more detail in the revised version of our manuscript.*

- Data on respiration is missing entirely.

*AR: Please see previous reply to comment.*

- The large variability of DIC upon the nutrient addition is overwhelming and not well explained, also not matching the trends in the other parameters. Provide better explanation.

*AR: The authors do not agree with this statement. As explained in the first paragraph of the discussion, the DIC decrease is due to primary productivity being stimulated by the nutrient addition. The fact that this is difficult to detect in other parameters than the nutrients themselves is that they are either hardly affected, i.e., TA, or only slightly, e.g., pH and pCO2, which are difficult to spot due to the rather large initial treatment differences as opposed to a more uniform DIC.*

- Where did you take the 95% for full equilibration from?

*AR: This is based on a simply forward calculation, assuming average gas exchange rates determined in our study and calculating how long it would take for a 95% equilibration. The 95% threshold was chosen to provide a more realistic estimate of the equilibration time, as the process follows an exponential pattern and reaching a true 100% is then virtually impossible.*

- No coccolithophore or diatom data presented??? It is literally impossible to draw some of the results and conclusion in this paper unless there is data available for this.

*AR: Coccolithophore data is available in the Supplement, FS 5. We will highlight this more prominently in the main text. For diatoms, we only have BSi data as a proxy that can be correlated to each treatment. It will be added in the appendix as well.*

- Are there any taxonomic or metagenomic assessments to resolve zooplankton community and why the decision on cutting it at 280um?

*AR: No taxonomic or metagenomic data is available to resolve RZ. We didn't decide on the cutting at 280um, this cut comes from the methodology applied in Marín-Samper et. al, 2024. 280 μm have been found to being a good compromise to not exclude too much of the natural community and at the same time ensure reproducibility between replicate incubations.*

**3. Results and Interpretation**

**Calcification:** Coccolithophore calcification followed an optimum curve relative to pCO2, with a peak around 250 μatm and suppression at extremes. However, in the figure S4a, calcification is below 0 for the two highest treatments, which is not explained anywhere in the text. Does this indicate dissolution. Even less severe treatments are still just hardly above 0, especially before the addition of nutrient part, which signifies lack of calcification overall, and only just happening in the first three treatments. How does this align with the NCP, can you correlate? And how does it align with the CALC, could it have any effect on the TA? Is this species-specific, could it be due to any other calcifiers (not just the autotrophs)? In general, the drawback of this is also that no other potential calcifiers have been implicated in the CALC, only the autotrophs. Are there any data available to support this, or discount for the impact of zooplankton on the CALC?

*AR: Having calcification hovering around the 0 line or being negative is most likely related to the inherent uncertainty stemming from a mass balance involving four measurements with their individual uncertainties (TA, salinity, nitrate and phosphate). However, the fact that we find an optimum curve suggests that despite these uncertainties, the overarching pattern is preserved when calculating cumulative calcification for the entire experiment. We will mention this in the discussion. Concerning CALC and NCP, both are derived considering changes in TA (see Eqs. 5 and 6). This in turn also means that calcification by any organisms is captured by this method. The fact, however, that the cumulative calcification correlated well with cumulative coccolithophore counts (Suppl. Fig. S5) suggests that they were the dominant calcifiers in our experiment.*

**Net Community Production (NCP):** NCP was significantly higher in silicate treatments post-fertilization, with no direct pCO2 effect. But the effect of the pH was not investigated and should be included in ANOVA. Also, why is NCP related to Si and not to Ca2+ treatment- again, data on diatoms and phytoplankton are absolutely essential, otherwise this all on the level of inferences.

*AR: Given the set-up of this experiment, pH and pCO$_2$ are intimately correlated, i.e., there is a quasi-linear relationship of proton concentration and pCO2. So, either one of them could have been chosen, but we decided to stick to the one parameter that is relevant to all aspects of the manuscript. We will add this piece of information. Concerning the Si/Ca question, there is a relatively large background of Ca$^{2+}$ in seawater, meaning that our additions change concentrations only by a few percent (0.8 – 3.1%). In contrast Si is a macronutrient needed by a particular group of producers and that changed in between the two treatments by several orders of magnitude. Hence, Si seems to be behind the mineral-effect on NCP rather than Ca. Even though we do not have data on diatoms, BSi data will also be added to the Supplement as a proxy.*

Also, how does Chla correlate with NCP and Calcification (Figure S4a-c and S1f)?

*AR: While there is a reasonable correlation between daily changes in NCP and chlorophyll standing stocks, as one would expect, the latter cannot be compared to calculated calcification, as what is shown here are cumulative changes not daily rates.*

**Zooplankton Respiration:** Respiration declined with decreasing pCO2 and was lower in Si treatments, but in general, this aspect is largely underexplored and insufficiently presented. Much more effort needs to be put in explaining respiration data and how it links to suggested trophic-level complexity. Present the data on respiration beyond 2 days, 2-day data is insufficient, compare the pre and post nutrient treatment.

*AR: Data on zooplankton respiration is cumulative. Hence, all values throughout the entire experiment are factored in when we take the cumulative mean for the last 2 days. Also, comparing RZ pre and post nutrient addition, it appears that there is a consistent trend. We will make this clearer in the text.*

The results of respiration are also fundamental in explaining some of the effects and should be put in the Results, not Discussion, section.

*AR: As explained previously, we consider zooplankton respiration to better fit in the discussion and would like to keep it there, as it resulted from comparing to and discussing a dataset from another publication. We will make the text clearer regarding this topic.*

**Discussion:**
In general, this study is really divided in two parts:
• Pre-nutrient treatment that is represented of the fjord environment under OAE and post-nutrient that is NO LONGER representative of the oligotrophic fjord conditions, whereby the used communities were not acclimated to such increases in nutrients and is just a mesocosm trial of OAE with nutrients. In such systems, the communities and species could react completely

differently than under such artificial conditions. This aspect is now touched upon in the results and discussion and I would like the authors to fully dedicate the effort on the potential confounding effects due to such nutrient addition and how different the fjord system response to OAE would be if such strong nutrient artificial addition was not present- Fco2 still high, but NPC insignificant, what about respiration etc?

*AR: Thank you for the suggestion. We will broaden the discussion on this topic and add supplementary figures on responses in the two phases as well as the whole experiment. We will also include a summary table (exemplified as follows).*

| Parameter | Phase 1 | Phase 2 | Phases 1+2 |
|---|---|---|---|
| cCALC | Gradient only, optimum curve | Gradient only, optimum curve | Gradient only, optimum curve |
| cNCP$_{DIC}$ | Gradient only | Mineral only | Mineral only |
| cRZ | Gradient only | None | Gradient and Mineral, no interaction |
| cBSi | Mineral only | Mineral only | Mineral only |

•      In addition, no evaluation of the longer-term dynamics to capture seasonal or successional effects is presented.

*AR: Extrapolating our six weeks results to obtain longer-term dynamics will be difficult as to the unknown of seasonal and successional variability.*

REFERENCES

Douglas, N. K., Byrne, R. H., Achieving accurate spectrophotometric pH measurements using unpurified meta-cresol purple, Mar. Chem., 190:66.72 (2017).

Ferderer, A., Schulz, K. G., Riebesell, U., Baker, K. G., Chase, Z., Bach, L. T., Investigating the effect of silicate- and calcium-based ocean alkalinity enhancement on diatom silicification, Biogeosci., 21:2777–2794 (2024).

Schulz, K. G., Riebesell, U., Bellerby, R. G. J., Biswas, H., Meyeröfer, M., Müller, M. N., Egge, J. K., Nejstgaard, J. C., Neill, C., Wohlers, J., Zöllner, E., Build-up and decline of organic matter during PeECE III, Biogeosci., 5:707–718 (2008).

Schulz, K. G., Bach, L. T., Bellerby, R. G. J., Bermúdez, R., Büdenbender, J., Boxhammer, T., Czerny, J., Engel, A., Ludwig, A., Meyerhöfer, M., Larsen, A., Paul, A. J., Sswat, M., Riebesell, U., Phytoplankton blooms at increasing levels of atmospheric carbon dioxide: Experimental evidence for negative effects on prymnesiophytes and positive on small picoeukaryotes, Frontiers, Mar. Sci., 4:64 (2017).